# Modeling the initial phase of COVID-19 epidemic: The role of age and disease severity in the Basque Country, Spain

**Akhil Kumar Srivasrav** [1☯], **Nico Stollenwerk** [1,3☯], **Joseba Bidaurrazaga Van-Dierdonck** [4], **Javier Mar** [5,6,7], **Oliver Ibarrondo** [5], **Maíra Aguiar** [1,2,3☯] *

**1** Basque Center for Applied Mathematics, BCAM, Bilbao, Spain, **2** Ikerbasque, Basque Foundation for Science, Bilbao, Spain, **3** Dipartimento di Matematica, Universita degli Studi di Trento, Trento, Italy, **4** Public Health, Basque Health Department, Bilbao, Spain, **5** Osakidetza Basque Health Service, Debagoiena Integrated Healthcare Organisation, Research Unit, Arrasate-Mondragón, Guipúzcoa, Spain, **6** Biodonostia Health Research Institute, Donostia-San Sebastián, Guipúzcoa, Spain, **7** Kronikgune Institute for Health Services Research, Economic Evaluation Unit, Barakaldo, Spain

☯ These authors contributed equally to this work.

* mairaguiar@bcamath.org

**Data Availability Statement:** All relevant data are available within the paper.

## Abstract

Declared a pandemic by the World Health Organization (WHO), COVID-19 has spread rapidly around the globe. With eventually substantial global underestimation of infection, by the end of March 2022, more than 470 million cases were confirmed, counting more than 6.1 million deaths worldwide. COVID-19 symptoms range from mild (or no) symptoms to severe illness, with disease severity and death occurring according to a hierarchy of risks, with age and pre-existing health conditions enhancing risks of disease severity. In order to understand the dynamics of disease severity during the initial phase of the pandemic, we propose a modeling framework stratifying the studied population into two groups, older and younger, assuming different risks for severe disease manifestation. The deterministic and the stochastic models are parametrized using epidemiological data for the Basque Country population referring to confirmed cases, hospitalizations and deaths, from February to the end of March 2020. Using similar parameter values, both models were able to describe well the existing data. A detailed sensitivity analysis was performed to identify the key parameters influencing the transmission dynamics of COVID-19 in the population. We observed that the population younger than 60 years old of age would contribute more to the overall force of infection than the older population, as opposed to the already existing age-structured models, opening new ways to understand the effect of population age on disease severity during the COVID-19 pandemic. With mild/asymptomatic cases significantly influencing the disease spreading and control, our findings support the vaccination strategy prioritising the most vulnerable individuals to reduce hospitalization and deaths, as well as the non-pharmaceutical intervention measures to reduce disease transmission.

**Funding:** M. A. has received funding from the European Union's Horizon 2020 research and innovation programme under the Marie Sklodowska-Curie grant agreement No 792494. A. K.S., N.S, J.B.V, J.M, O.I. and M.A received funding from he Basque Government through the "Mathematical Modeling Applied to Health" Project. A.K.S, N.S and M.A, have received funding from the Basque Government through the BERC 2022-2025 program and by Spanish Ministry of Sciences, Innovation and Universities: BCAM Severo Ochoa accreditation SEV-2017-0718. The funders had no role in study design, data collection and analysis, decision to publish, or preparation of the manuscript.

**Competing interests:** The authors have declared that no competing interests exist.

# 1 Introduction

More than two years have passed since COVID-19, a severe respiratory syndrome caused by a new coronavirus, was identified by the Chinese authorities in January 2020 [1]. Declared a global pandemic by the World Health Organization (WHO) in March 2020 [2], COVID-19 symptoms range from asymptomatic/mild to severe illness, with age and pre-existing health conditions increasing the likelihood of disease severity [3]. Vaccines against COVID-19 have been developed in record time and are now globally distributed [4, 5]. Although these vaccines are remarkably effective against severe disease, the so called sterilizing immunity, occurring when vaccinated individuals cannot transmit the virus, is still being evaluated.

Based on previous research experiences applied to other infectious diseases [6–14], and more recently applied to COVID-19 dynamics [15–19], the role of asymptomatic infections have been studied, showing that vaccine performance is driven by the ability of asymptomatic or mild disease cases transmitting the virus, with an eventual increase on the number of overall infections in a population [20, 21].

As an example of the pandemic's impact in Europe, Spain has reported, by the end of March 2022, more than 11.5 million COVID-19 cases and over 100 thousand deaths [22, 23], with a significantly higher mortality rate for individuals older than 65 years of age [24, 25], in agreement with what was also observed in different European countries [26].

As the COVID-19 pandemic progressed, task forces have been created to assist public health managers and governments during the COVID-19 crisis, and research on mathematical modeling became critical to understand the epidemiological dynamics of COVID-19. Modeling studies to evaluate COVID-19 dynamics worldwide have been widely published. Using both, deterministic and stochastic approaches, models were developed to investigate disease spreading in different epidemiological contexts as well as the impact of the control measures so far implemented. Using the existing empirical data, these models have given insights on disease transmission rates, the effect of quarantine or use of facial masks, for example, with modeling assumptions statistically tested with the available empirical data [27–30].

Within the COVID-19 Basque Modeling Task Force (BMTF), a flexible stochastic framework was developed to describe the epidemics in terms of disease spreading and control in the Basque Country, Spain, giving projections on the national health system needs over time. The SHARUCD framework was parameterized and validated with epidemiological data continuously collected and provided by the Basque Health Department and the Basque Health Service (Osakidetza), and has been used, up to date, to monitor COVID-19 spreading and control over the course of the pandemic [15–21]. Model refinements and results on the evolution of the epidemics in the Basque Country are updated on a monthly basis and are publicly available as an online dashboard [5].

As a continuation of the BMTF efforts, we developed an age-stratified mathematical model framework to understand the epidemiological dynamics of COVID-19 introduction phase in the Basque Country. The models are calibrated with the available data referring to confirmed cases, hospitalizations and deaths, from February to the end of March 2020, in the Basque Country, prior to any intervention measure. After a careful data analysis, the population was divided into two groups, namely young and old. As opposed to the existing age structured models suggesting higher infection rate for individuals older than 60 years of age [31–34] than for younger individuals, our modeling assumption implies that while the risk for developing severe disease is higher for the older population, disease transmission is significantly driven by the mobile younger population.

A detailed sensitivity analysis was performed to identify the key parameters influencing the transmission dynamics of COVID-19 in the population, opening new ways to understand the effect of age on disease severity during the pandemic. In terms of policy implications, our findings support the vaccination strategy prioritising the most vulnerable individuals [16], particularly to reduce hospitalization and deaths [21], as well as the non-pharmaceutical intervention measures that are still advised by the WHO to reduce disease transmission.

This paper is organized as follows. Section 2 presents the deterministic and the stochastic models formulation, followed by the model analysis. Section 3 is dedicated to data analysis, model calibration and parameter estimation. In Section 4 we present the models simulation and results, including a detailed sensitivity analysis for the parameters involved in reproduction number. We conclude this work with a discussion on the results obtained by both modeling approaches.

## 2 Materials and methods

Using age stratified data for COVID-19 incidences for tested positive cases, hospitalizations and deaths in the the Basque Country, this work is applied to the initial phase of the pandemic. Using statistical tools to analyse these data, we define as severe cases all hospitalized individuals, including the intensive care unit (ICU) admissions, for young ($H_1$) and old patients ($H_2$), reported from February 15 to March 25, 2020. It is important to mention that at the beginning of the pandemic, due to the testing capacity limitations, only patients with severe symptoms were tested using the PCR (Polymerase Chain Reaction) method.

### 2.1 The deterministic model

This model framework is a refinement of the model proposed by Srivastav et al. [27, 35]. For both age groups, young and old, susceptible individuals become exposed and infected $E_1(t)$ and $E_2(t)$, developing either mild/asymptomatic $A_1(t)$ and $A_2(t)$ or severe/hospitalized $H_1(t)$ and $H_2(t)$ disease. While mild/asymptomatic infections are assumed to recover, severe disease might evolve to death $D$. The parameter $\phi$ differentiates the disease transmission between hospitalized ($H_1 + H_2$) and mild/asymptomatic infections ($A_1 + A_2$), and the parameter $\epsilon$ is introduced to differentiate the infectivity of asymptomatic young individuals ($A_1$) with respect to the baseline infectivity for the elderly individuals $A_2(t)$ in the Basque Country population of $N = 2.6$ million individuals.

The seriousness of symptoms from viral infections is often correlated with the amount of the virus in the body [36, 37]. Justified by the differences observed in viral load during the COVID-19 infection, lower for mild/asymptomatic and higher for severe/hospitalized cases, we assume $\epsilon < 1$, indicating that young individuals have smaller infectivity than the elderly individuals. This assumption relies on the epidemiological observation of young individuals developing mild or no symptoms during the infection as opposed to the observation of severe symptoms occurring mostly in older ages, shaping the disease transmissibility pattern in a population. The parameter $\phi$ is a scaling factor used to differentiate the infectivity of mild/asymptomatic infections ($\phi\beta$) with respect to the baseline infectivity of severe/hospitalized cases ($\beta$). The value of $\phi$ can be tuned to reflect different situations: a value of $\phi < 1$ reflects the fact that severe cases have larger infectivity than mild cases (e.g., due to enhanced coughing and sneezing), while $\phi > 1$ indicates that asymptomatic individuals and mild cases contribute more to the spread of the infection (e.g., due to their higher mobility and possibility of interaction) than the severe cases which are more likely to be detected and isolated [15]. Here, we assume

$\phi > 1$, with asymptomatic individuals contributing more to the force of infection than the hospitalized individuals [38, 39].

The total population $N$ is divided into ten compartments, stratified into two age groups, young and old. Susceptible $S_1(t)$ and $S_2(t)$, Exposed $E_1(t)$ and $E_2(t)$, mild/Asymptomatic $A_1(t)$ and $A_2(t)$ or severe/Hospitalized $H_1(t)$ and $H_2(t)$ cases. Labels 1 and 2 refers to the young and to the old age populations respectively. Two extra classes to accommodate individuals from both age-groups are also considered. The deceased class $D(t)$, for those who died from COVID-19, and finally the recovered class $R(t)$, counting all individuals recovered from the disease.

For the mathematical modelling framework development, we make the following assumptions:

1. The total population $N$ is constant.

2. The susceptible young individuals $S_1$ become exposed to the infection $E_1$ by contacting infectious individuals $A_1$, $A_2$ and $H_1$, $H_2$ at rates $\phi\beta$ and $\beta$, respectively.

3. The susceptible old individuals $S_2$ become exposed to the infection $E_2$ by contacting infectious individuals $A_1$, $A_2$ and $H_1$, $H_2$ at rates $\phi\beta$ and $\beta$, respectively.

4. With $i$ = 1, 2, for young and old respectively, exposed individuals $E_i$ will develop mild/asymptomatic infection $A_i$ with rate $a\eta_i$ while the remaining individuals developing severe symptoms will be admitted to a hospital facility $H_i$ with rate $(1 - a)\eta_i$.

5. While young and old asymptomatic individuals recover from COVID-19 infection ($R$) with rate $\alpha_1$ and $\alpha_3$ respectively, hospitalized young individuals will recover with rate $\alpha_2$ while hospitalized old individuals will recover with rate $\alpha_4$. Young and old hospitalized individuals will eventually die ($D$) with rate $\delta_1$ and $\delta_2$ respectively. The description of model framework parameters can be found in Table 1

The flow diagram for the disease related stages of our proposed model is shown in Fig 1, which translates into the following ODE system describing the temporal evolution of the

**Table 1. Description of model framework parameters.**

| Parameter | Description |
|---|---|
| $\beta$: | baseline COVID-19 transmission rate |
| $\phi$: | scaling factor used to differentiate the infectivity of severe/hospitalized cases |
| $\epsilon$: | scaling factor used to differentiate the infectivity of young and elderly mild/asymptomatic cases |
| $\delta_1$: | disease induced death rate for hospitalized young individuals |
| $\delta_2$: | disease induced death rate for hospitalized old individuals |
| $\eta_1$: | hospitalization rate for young individuals |
| $\eta_2$: | hospitalization rate for old individuals |
| $\alpha_1$: | recovery rate of asymptomatic young individuals |
| $\alpha_3$: | recovery rate of asymptomatic old individuals |
| $\alpha_2$: | recovery rate of hospitalized young individuals |
| $\alpha_4$: | recovery rate of asymptomatic old individuals |
| $a$: | Fraction of exposed population developing mild/asymptomatic disease |
| $(1 - a)$: | Fraction of exposed population developing severe/hospitalized disease |

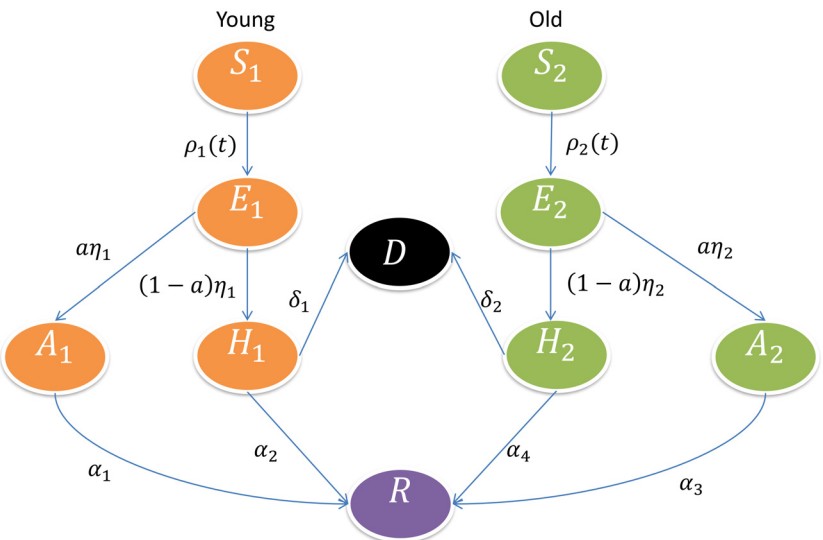

**Fig 1. With $\rho_1(t) = \beta S_1[\phi(A_1 + \epsilon A_2) + (H_1 + H_2)]$ and $\rho_2(t) = \beta S_2[\phi(A_1 + \epsilon A_2) + (H_1 + H_2)]$, disease related stages are shown in orange color for young population and in light green for the old population.** Deceased and recovered population include both age groups and are shown in black and purple, respectively.

number of individuals in each of the model compartments:

$$
\begin{aligned}
\frac{dS_1}{dt} &= -\beta S_1[\phi\{A_1 + \epsilon A_2\} + (H_1 + H_2)] \\
\frac{dE_1}{dt} &= \beta S_1[\phi\{A_1 + \epsilon A_2\} + (H_1 + H_2)] - \eta_1 E_1 \\
\frac{dA_1}{dt} &= a\eta_1 E_1 - \alpha_1 A_1 \\
\frac{dH_1}{dt} &= (1-a)\eta_1 E_1 - \delta_1 H_1 - \alpha_2 H_1 \\
\frac{dS_2}{dt} &= -\beta S_2[\phi\{A_1 + \epsilon A_2\} + (H_1 + H_2)] \\
\frac{dE_2}{dt} &= \beta S_2[\phi\{A_1 + \epsilon A_2\} + (H_1 + H_2)] - \eta_2 E_2 \\
\frac{dA_2}{dt} &= a\eta_2 E_2 - \alpha_3 A_2 \\
\frac{dH_2}{dt} &= (1-a)\eta_2 E_2 - \delta_2 H_2 - \alpha_4 H_2 \\
\frac{dR}{dt} &= \alpha_1 A_1 + \alpha_2 H_1 + \alpha_3 A_2 + \alpha_4 H_2. \\
\frac{dD}{dt} &= \delta_1 H_1 + \delta_2 H_2
\end{aligned}
\tag{1}
$$

## 2.2 Existence of equilibrium points and the basic reproduction number ($R_0$)

While the disease-free equilibrium of the system is given by $E_0 = (S_1^0, E_1^0, A_1^0, H_1^0, S_2^0, E_2^0, A_2^0, H_2^0, R^0, D^0) = (N_1^0, 0, 0, 0, 0, N_2^0, 0, 0, 0, 0)$, the basic

reproduction number $R_0$ can be found by using the next generation matrix method [40], and is given by:

$$R_0 = \beta S_1^0 \left\{ \frac{(1-a)}{(\delta_1 + \alpha_2)} + \frac{\phi a}{\alpha_1} \right\} + \beta S_2^0 \left\{ \frac{(1-a)}{(\delta_2 + \alpha_4)} + \frac{\phi \epsilon a}{\alpha_3} \right\}$$

The quantity $R_1 = \beta S_1^0 \left\{ \frac{(1-a)}{(\delta_1 + \alpha_2)} + \frac{\phi a}{\alpha_1} \right\}$ is defined for the young group population and the quantity $R_2 = \beta S_2^0 \left\{ \frac{(1-a)}{(\delta_2 + \alpha_4)} + \frac{\phi \epsilon a}{\alpha_3} \right\}$ is defined for the old group population. The quantity $R_0 = R_1 + R_2$ is the average number of secondary cases produced in a completely susceptible population by an index case, during the infectious period.

The calculation of the basic reproduction number $R_0$ is shown in the S1 File. We can summarize our findings in the following theorems.

**Theorem 1.1** *If $R_0 < 1$, the disease-free equilibrium $E_0 = (N_1^0, 0, 0, 0, N_2^0, 0, 0, 0, 0, 0)$ of the system (1) is locally asymptotically stable, and if $R_0 > 1$, the disease-free equilibrium $E_0$ is unstable.*

Next, we state globally asymptotically stability of disease-free equilibrium.

**Theorem 1.2** *If $R_0 < 1$, the disease-free equilibrium $E_0 = (N_1^0, 0, 0, 0, N_2^0, 0, 0, 0, 0, 0)$ of the system (1) is globally asymptotically stable whenever eigenvalue of the matrix $F - V$ are having negative real parts, and if $R_0 > 1$, the disease-free equilibrium $E^0$ is unstable* [41].

The proof of the global stability can be found in the S1 File.

## 2.3 The stochastic model

As all natural systems are prone to stochastic fluctuations, we extended our deterministic model, see Equation System 1, to the corresponding stochastic model. The derivation of the stochastic model and its analysis are important when populations are small, and hence with the dynamics being severely affected by small changes in the parameter values. Thus, for the initial phase of the COVID-19 outbreak, the stochastic model setup is the most appropriate modeling approach to be used for a local epidemiological evaluation.

The derivation of a stochastic differential equation (SDE) model is a diffusion approximation from the underlying state discrete Markov process [17, 41–45]. Let

$$X(t) = (X_1(t), X_2(t), X_3(t), X_4(t), X_5(t), X_6(t), X_7(t), X_8(t), X_9(t), X_{10}(t))^T$$

be a continuous random variable for

$$[S_1(t), E_1(t), A_1(t), H_1(t), S_2(t), E_2(t), A_2(t), H_2(t), R(t), D(t)]^T,$$

where $T$ denotes transpose of the matrix. Further, let $\Delta X = X(t + \Delta t) - X(t) = (\Delta X_1, \Delta X_2, \Delta X_3, \Delta X_4 \ldots)^T$ denotes the random vector for the change in random variables during time interval $\Delta t$. Here, we write the transition maps which define all possible changes between disease states in the SDE model. State changes and their probabilities are presented in Table 2, followed by

**Table 2. Possible changes of states and their probabilities.**

| Possible state change | Probability of state change |
|---|---|
| $(\Delta X)_1 = (-1, 1, 0, 0, 0, 0, 0, 0, 0, 0)^T$ <br> Change when young susceptible meet infected individuals and move to the young exposed class | $P_1 = \beta X_1 [\phi(X_3 + \epsilon X_7) + (X_4 + X_8)]\Delta t + O(\Delta t)$ |
| $(\Delta X)_2 = (0, -1, 1, 0, 0, 0, 0, 0, 0, 0)^T$ <br> Change when fraction of young exposed become infectious and move to the young asymptomatic infected class | $P_2 = a\eta_1 X_2 \Delta t + O(\Delta t)$ |
| $(\Delta X)_3 = (0, -1, 0, 1, 0, 0, 0, 0, 0, 0)^T$ <br> Change when fraction of young exposed become infectious and move to the young hospitalized class | $P_3 = (1-a)\eta_1 X_2 \Delta t + O(\Delta t)$ |
| $(\Delta X)_4 = (0, 0, -1, 0, 0, 0, 0, 0, 1, 0)^T$ <br> Change when young asymptomatic infected recovers and move to the recovered class | $P_4 = \alpha_1 X_3 \Delta t + O(\Delta t)$ |
| $(\Delta X)_5 = (0, 0, 0, -1, 0, 0, 0, 0, 0, 1)^T$ <br> Change when young hospitalized die and move to the deceased class | $P_5 = \delta_1 X_4 \Delta t + O(\Delta t)$ |
| $(\Delta X)_6 = (0, 0, 0, -1, 0, 0, 0, 0, 1, 0)^T$ <br> Change when young hospitalized individuals recover and move the recovered class | $P_6 = \alpha_2 X_4 \Delta t + O(\Delta t)$ |
| $(\Delta X)_7 = (0, 0, 0, 0, -1, 1, 0, 0, 0, 0)^T$ <br> Change when old susceptible meet infected individual and move to the old exposed class | $P_7 = \beta X_1 [\phi(X_3 + \epsilon X_7) + (X_4 + X_8)]\Delta t + O(\Delta t)$ |
| $(\Delta X)_8 = (0, 0, 0, 0, 0, -1, 1, 0, 0, 0)^T$ <br> Change when fraction of old exposed become infectious and move to the old asymptomatic infected class | $P_8 = a\eta_2 X_6 \Delta t + O(\Delta t)$ |
| $(\Delta X)_9 = (0, 0, 0, 0, 0, -1, 0, 1, 0, 0)^T$ <br> Change when fraction of old exposed become infectious and move to the old hospitalized class | $P_9 = (1-a)\eta_2 X_6 \Delta t + O(\Delta t)$ |
| $(\Delta X)_{10} = (0, 0, 0, 0, 0, 0, -1, 0, 1, 0)^T$ <br> Change when old asymptomatic infected recovers and move to the recovered class | $P_{10} = \alpha_3 X_7 \Delta t + O(\Delta t)$ |
| $(\Delta X)_{11} = (0, 0, 0, 0, 0, 0, 0, -1, 0, 1)^T$ <br> Change when old hospitalized die and move to the deceased class | $P_{11} = \delta_2 X_8 \Delta t + O(\Delta t)$ |
| $(\Delta X)_{12} = (0, 0, 0, 0, 0, 0, 0, -1, 1, 0)^T$ <br> Change when old hospitalized individuals recover and move to the recovered class | $P_{12} = \alpha_4 X_8 \Delta t + O(\Delta t)$ |
| $(\Delta X)_{13} = (0, 0, 0, 0, 0, 0, 0, 0, 0, 0)^T$ <br> No change | $P_{13} = 1 - \sum_{i=1}^{12} \Delta t + O(\Delta t)$ |

the full SDE system 2.

$$
\begin{aligned}
dS_1 &= (-\beta S_1[\phi\{A_1 + \epsilon A_2\} + (H_1 + H_2)])dt - \sqrt{\beta S_1[\phi\{A_1 + \epsilon A_2\} + (H_1 + H_2)]}dW_1, \\
dE_1 &= [\beta S_1[\phi\{A_1 + \epsilon A_2\} + (H_1 + H_2)] - a\eta_1 E_1 - (1-a)\eta_1 E_1]dt \\
&\quad + \sqrt{\beta S_1[\phi\{A_1 + \epsilon A_2\} + (H_1 + H_2)}dW_1 - \sqrt{a\eta_1 E_1}dW_2 - \sqrt{(1-a)\eta_1 E_1}dW_3 \\
dA_1 &= [a\eta_1 E_1 - \alpha_1 A_1]dt + \sqrt{a\eta_1 E_1}dW_2 - \sqrt{\alpha_1 A_1}dW_4 \\
dH_1 &= [(1-a)\eta_1 E_1 - \delta_1 H_1 - \alpha_2 H_1]dt + \sqrt{(1-a)\eta_1 E_1}dW_3 - \sqrt{\delta_1 H_1}dW_5 - \sqrt{\alpha_2 H_1}dW_6 \\
dS_2 &= (-\beta S_2[\phi\{A_1 + \epsilon A_2\} + (H_1 + H_2)])dt - \sqrt{\beta S_2[\phi\{A_1 + \epsilon A_2\} + (H_1 + H_2)]}dW_7, \\
dE_2 &= [\beta S_2[\phi\{A_1 + \epsilon A_2\} + (H_1 + H_2)] - a\eta_2 E_2 - (1-a)\eta_2 E_2]dt \\
&\quad + \sqrt{\beta S_2[\phi\{A_1 + \epsilon A_2\} + (H_1 + H_2)}dW_7 - \sqrt{a\eta_2 E_2}dW_8 - \sqrt{(1-a)\eta_2 E_2}dW_9 \\
dA_2 &= [a\eta_2 E_2 - \alpha_3 A_2]dt + \sqrt{a\eta_2 E_2}dW_8 - \sqrt{\alpha_3 A_2}dW_{10} \\
dH_2 &= [(1-a)\eta_2 E_2 - \delta_2 H_2 - \alpha_4 H_4]dt + \sqrt{(1-a)\eta_2 E_2}dW_9 - \sqrt{\delta_2 H_2}dW_{11} \\
&\quad - \sqrt{\alpha_4 H_2}dW_{12} \\
dR &= [\alpha_1 A_1 + \alpha_2 H_1 + \alpha_3 A_2 + \alpha_4 H_2]dt + \sqrt{\alpha_1 A_1}dW_4 + \sqrt{\alpha_2 H_1}dW_6 + \sqrt{\alpha_3 A_2}dW_{10} \\
&\quad + \sqrt{\alpha_4 H_2}dW_{12} \\
dD &= [\delta_1 H_1 + \delta_2 H_2]dt + \sqrt{\delta_1 H_1}dW_5 + \sqrt{\delta_2 H_2}dW_{11}
\end{aligned}
\tag{2}
$$

**Table 3. Cumulative disease cases by age in the Basque Country.**

| | COVID-19 epidemiological data, from February 15 to March 25, 2020 | | | | | |
|---|---|---|---|---|---|---|
| | Raw | | | Normalized by $10^5$ people | | |
| age classes | positive cases | hospital admissions | deceased cases | positive cases | hospital admissions | deceased cases |
| 0–9 | 19 | 3 | 0 | 10 | 2 | 0 |
| 10–19 | 34 | 5 | 0 | 17 | 3 | 0 |
| 20–29 | 188 | 34 | 1 | 97 | 18 | 1 |
| 30–39 | 388 | 118 | 2 | 146 | 45 | 1 |
| 40–49 | 600 | 255 | 4 | 168 | 71 | 1 |
| 50–59 | 796 | 393 | 6 | 230 | 118 | 2 |
| 60–69 | 714 | 518 | 20 | 263 | 191 | 8 |
| 70–79 | 638 | 622 | 44 | 316 | 308 | 22 |
| 80+ | 680 | 523 | 146 | 432 | 332 | 93 |

The detailed derivation of the stochastic model can be found in the S1 File of this manuscript.

## 3 Data analysis and parameter estimation

### 3.1 Epidemiological data

Epidemiological data used in this study are provided by the Basque Health Department and the Basque Health Service (Osakidetza), continually collected with specific inclusion.

By March 4, 2022, around 600,000 cases were confirmed, with 32087 hospital admissions and 8788 deaths in the Basque Country. For the proposed model, the age stratification was decided after a careful data inspection and data fitting, followed by the parameter estimation.

We use the epidemiological data referring to the cumulative incidences of confirmed positive cases, hospitalizations, including ICU admissions, and deceased cases distributed by age groups available for the initial phase of the COVID-19 in the Basque Country, from February 15 to to March 25, 2020, as shown in Table 3.

Note that during this period, testing capacity was limited and therefore the positive detected cases were restricted to symptomatic individuals and eventually to their close contacts during the process tracing and testing strategy.

### 3.2 Model calibration method

Using MATLAB software, parameter estimation was performed using nonlinear least square method [47]. In detail, we search for the set of parameters $\hat{\Theta} = (\hat{\theta}_1, \hat{\theta}_2, \hat{\theta}_3 \ldots \hat{\theta}_n)$ that minimizes the sum of squared differences between the observed data $y_{t_i} = (y_{t_1}, y_{t_2} \ldots y_{t_n})$ and the corresponding model solution denoted by $(f(t_i, \Theta)$

$$\hat{\Theta} = argmin \sum_{i=1}^{n} \left( f(t_i, \Theta) - y_{t_i} \right)^2.$$

The Root Mean Square Error (RMSE) values for the deterministic and stochastic models are calculate using the following formula,

$$RMSE = \sqrt{\frac{1}{n} \sum_{i=1}^{n} \left( f(t_i, \Theta) - y_{t_i} \right)^2},$$

where $t_i$ are the time points at which the time series data are observed, and $n$ is the number of data points available for parameter inference. Hence, the model solution $f(t_i, \Theta)$ yields the best-fit to the time series data $y_{t_i}$.

### 3.3 Raw data and model fitting

These raw data distribution by age groups are shown in Fig 2.

During the initial phase of the pandemic, a strong correlation of positive cases and severe disease leading to hospitalizations is observed, see y-axis of Fig 2a) and 2b). Increased age appears to be a strong risk factor for developing severe illness with COVID-19 infections,

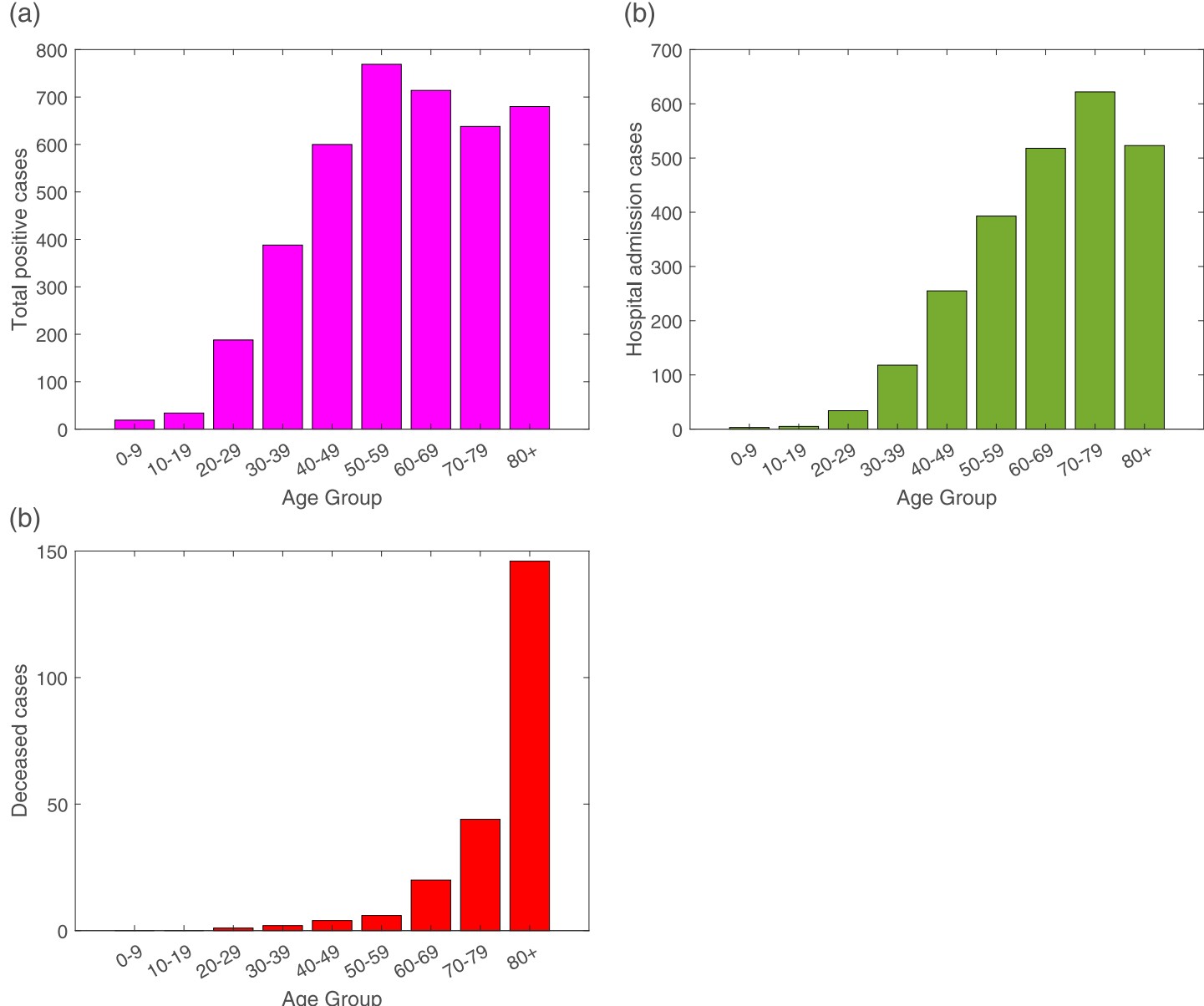

**Fig 2. From February 15 to March 25, 2020, raw data distribution for (a) total positive cases, (b) Hospital admission including ICU cases and (c) deceased cases.**

however, by looking at the raw data referring to the hospital admissions, this consideration is not very clear, with similar high hospitalization rates for individuals younger than 50 years of age and individuals older than 70 years of age. Nevertheless, when looking at the deceased cases, it is indeed observed that older adults have higher risk of severe outcomes. With potential underlying health conditions [3], most of deaths occurred in those older than 70 year of age, see Fig 2c).

Aiming to understand the role of population age heterogeneity on disease transmission and severe outcomes in the absence of vaccines and other non-pharmaceutical interventions, we consider the information obtained from the raw hospitalization data for the initial phase of the pandemic. While the young group includes individuals between 0–39 years of age, the old group considers the remaining individuals in the population older than 40 years of age. Models are calibrated with the data and the parameters reflecting the differences in disease transmission by age group are estimated.

The available data referring to cumulative hospital admission cases, for the young and the old age groups, are matched with both models, deterministic and stochastic. Fig 3 shows the models fitting to the empirical raw data. In this data matching scenario, the RMSE values for the deterministic and stochastic models are 0.55 and 0.47 respectively, indicating that the stochastic modeling approach explains better the existing data. The scaling factor parameter used to differentiate the infectivity of severe/hospitalized cases $\phi$, and scaling factor parameter used to differentiate the infectivity of young and elderly mild/asymptomatic cases $\epsilon$, were estimated to be

$$\phi = 1.2, \epsilon = 0.25$$

for the young group, and

$$\phi = 1.55, \epsilon = 0.4$$

for the old group. The other parameter values are fixed as suggested in [17]. Referring to the raw data, the used parameter values for the data fitting are listed in Table 4.

## 3.4 Normalized data and model fitting

The normalized raw data relative to the population size for each age class in the Basque Country is shown Table 3, with its visual age distribution shown in Fig 4.

Similarly to what was observed with the raw data, positive cases are increasing with age. The large majority of the deceased cases have been reported for the 80 years and older population group, confirming the strong correlation of severe disease outcome and age. Nevertheless, the normalization of the raw data shows clearer an increase of hospitalization rates for older age classes, allowing us to modify our modeling age stratification definition for young and old groups.

We summarize the distribution of disease cases using box plots to represent the deviation in the reported cases by age, see Fig 5, with the median being the measure of central tendency of the underlying distribution of the data as shown on Table 3.

Fig 5a) shows similar median values for individuals of 30 years and older, suggesting that they are more likely to develop symptoms than individuals at younger ages. In respect to the hospitalizations, see Fig 5b), the median values are similar for the individuals older than 50 years of age, suggesting that infections within these age groups are likely to be more severe requiring hospitalizations than for the younger ages, with individuals older than 80 years of age more likely to die from COVID-19 infection than any other age class, see Fig 5c). For these data, the age distribution assumption is now modified, considering individuals between 0–69

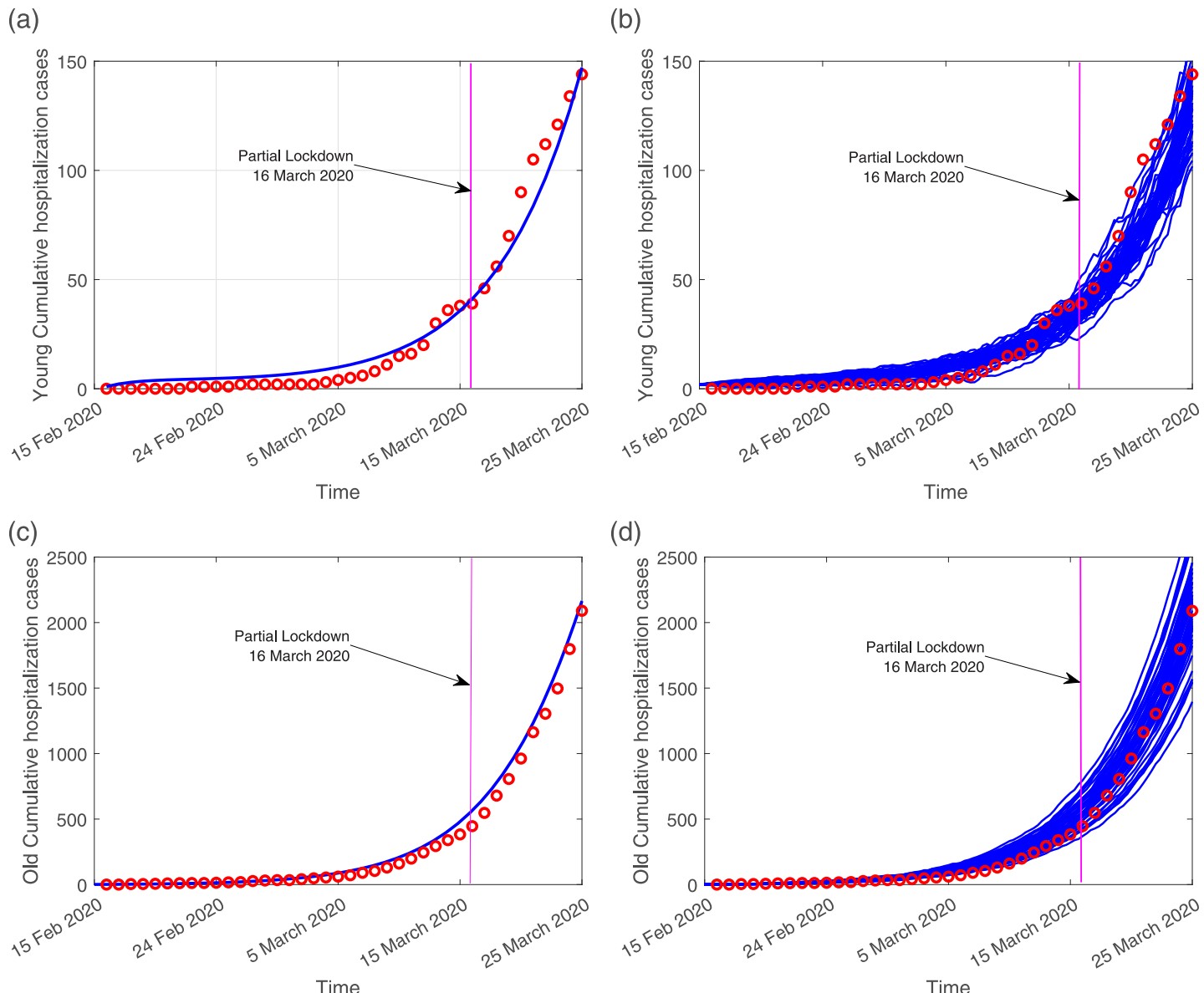

**Fig 3. On the left hand side, the deterministic model curve (blue line) and on the right hand side, the stochastic model realizations (in blue), fitting the cumulative empirical data referring to hospital admissions (red dots).** In (a) and in (b) data matching with model simulations for the young (0–39 years of age) age group. In (c) and (d) data matching with model simulations for the old (40 years and older) age group.

years of age as part of the young group and individuals older than 70 years of age as part of the old age group.

The cumulative empirical data for both age groups are matched with the deterministic system 1 and stochastic system 2 model simulations, see Fig 6.

The estimated values for the scaling factors used to differentiate the infectivity within the population are slightly smaller than the values obtained with the raw data. With

$$\phi = 1.5, \epsilon = 0.3$$

**Table 4. Parameters values used for model calibration.**

| Parameter | Normalized data values (fitting) | Raw data values (fitting) | Ref. |
|---|---|---|---|
| $\beta$: | 0.15 | 0.15 | [17] |
| $\phi$ (young): | 1.5 [1–2] | 1.2 [1–2] | fitted |
| $\epsilon$ (young): | 0.3 [0–1] | 0.25 [0–1] | fitted |
| $\phi$ (old): | 1.3 [1–2] | 1.55 [1–2] | fitted |
| $\epsilon$ (old): | 0.2 [0–1] | 0.4 [0–1] | fitted |
| $\delta_1$: | 0.003 [0.001–0.004] | 0.0012 [0.001–0.004] | fitted |
| $\delta_2$: | 0.04 [0.02–0.05] | 0.025 [0.02–0.05] | fitted |
| $\eta_1$: | 0.035 [0.0–0.5] | 0.035 [0.0–0.5] | [17] |
| $\eta_2$: | 0.03 [0.0–0.05] | 0.03 [0.0–0.05] | [17] |
| $\alpha_1$: | 0.02 [0.0–0.09] | 0.02 [0.0–0.09] | [17] |
| $\alpha_3$: | 0.05 [0.0–0.09] | 0.05 [0.0–0.09] | [17] |
| $\alpha_2$: | 0.01 [0.0–0.09] | 0.01 [0.0–0.09] | [17] |
| $\alpha_4$: | 0.03 [0.0–0.09] | 0.03 [0.0–0.09] | [17] |
| $a$: | 0.02 | 0.02 | [17] |

for the young population, and

$$\phi = 1.3, \epsilon = 0.2$$

for the old population, the disease induced death rate for hospitalized young and old groups, $\delta_i$, are also estimated. Referring to the normalized data, the model parameters used for fitting the data are shown in Table 4. This parameter set will be used in the further sections of this manuscript.

The RMSE values were calculated to be 0.35 and 0.2 for the deterministic and stochastic models respectively. With lower values than the values obtained by fitting the raw data, again, the stochastic model has a better fitting (with a lower RMSE value than the deterministic model), confirming that the stochastic approach explains better the existing normalized data.

# 4 Results

## 4.1 Sensitivity analysis

A detailed sensitivity analysis is performed to determine how the parameter values variation will affect the reproduction number ($R_0$) of the system. These results are of use to guide public health authorities during a disease outbreak.

In order to detect which are the parameters with higher impact on the $R_0$ measure, with effects to increase or to decrease its value and consequently to define which parameters are to be targeted by intervention measures, we use the the normalized forward sensitivity method index of a variable to a parameter [48]. The normalized forward sensitivity index of $R_0$ is defined using partial derivatives, showing the variation of the variable with respect to a given parameter $p$, as follows

$$\gamma_p^{R_0} = \frac{\partial R_0}{\partial p} \frac{p}{R_0}.$$

While the magnitude of the $R_0$ measure increases as the values of $\beta$, $a$, $\phi$, and $\epsilon$ parameters increase (positive indices), an inverse relation with the $R_0$ value is observed for the $\delta_1$, $\alpha_1$, $\alpha_2$, $\alpha_3$, $\alpha_4$, and $\delta_2$ parameters, with negative indices, i.e., as the parameter values increase, the

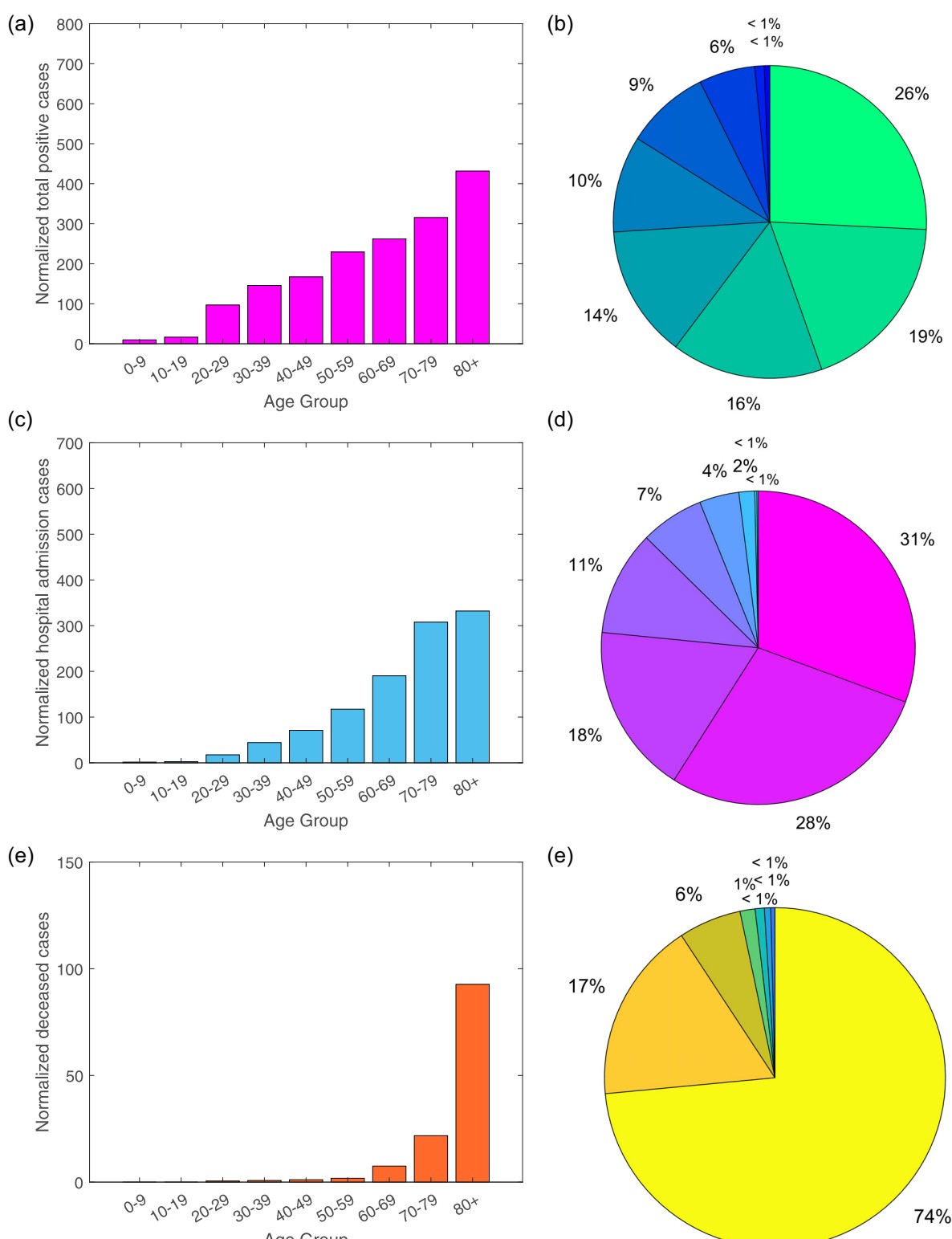

**Fig 4. From February 15 to March 25, 2020, normalized data distribution by age group.** The data is presented as confirmed cases per 100000 people. In (a-b) total positive cases, (c-d) hospitalized cases and (e-f) deceased cases.

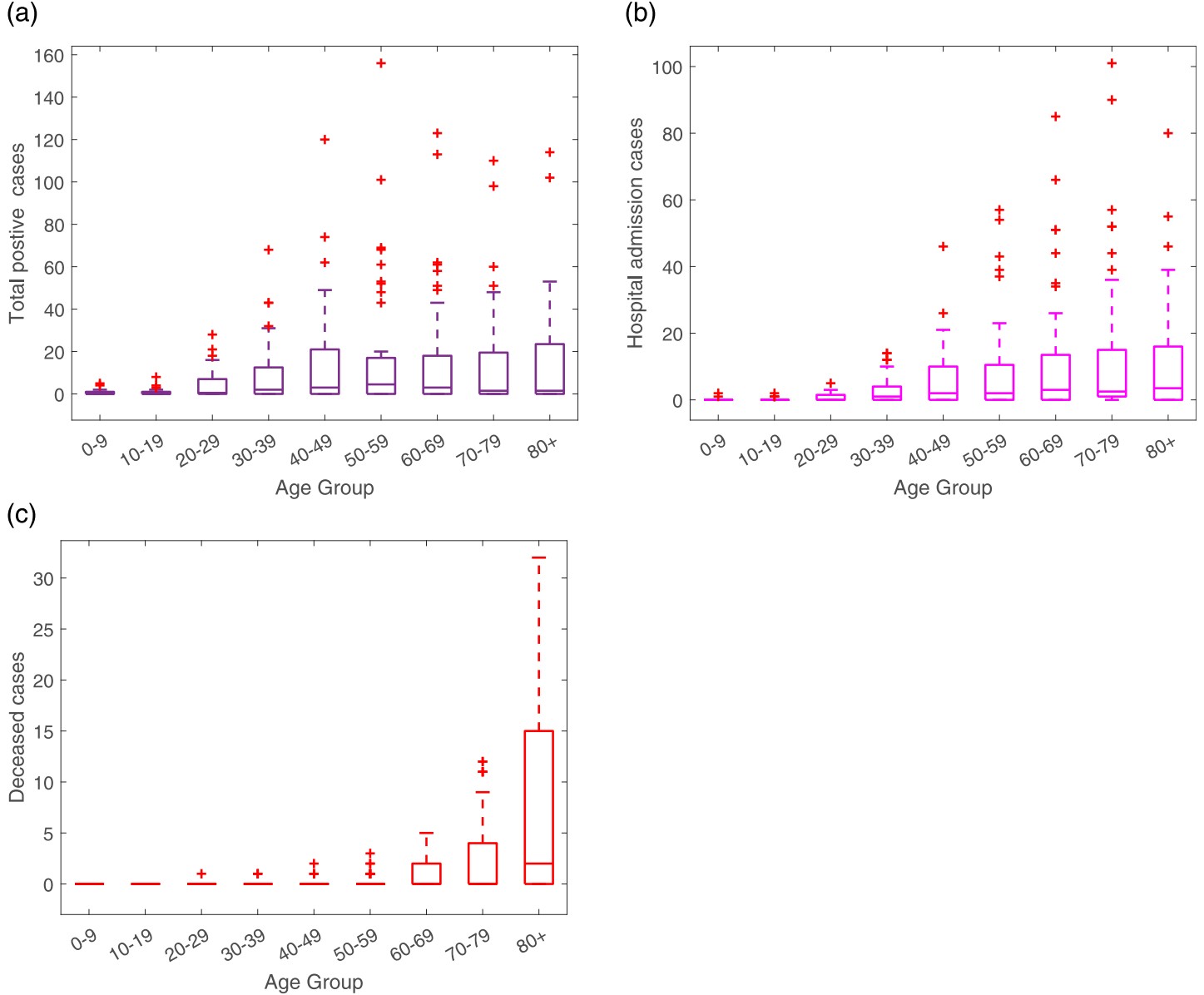

**Fig 5. Box plots for (a) total positive cases, (b) hospitalized cases and (c) deceased cases.** Horizontal lines denote lower quartile, median and upper quartile, with dots showing outliers.

magnitude of $R_0$ decreases. The sensitivity index of $R_0$ for the parameter $\beta$ is 1, meaning that $R_0$ increases or decreases with the same percentage as the parameter $\beta$ varies, see Fig 7.

Complementary to the forward sensitivity method index analysis above, we use the spline regression method to fit 10000 points for a range of each parameter value. The quantification of the parameter variation effect on the $R_0$ value is shown in Fig 8, confirming that the increase of the transmission rate $\beta$, the fraction of asymptomatic individuals $a$, and the scaling factors differentiating the disease transmission withing the population, $\phi$ and $\epsilon$, values affects significantly the behaviour of the $R_0$ measure.

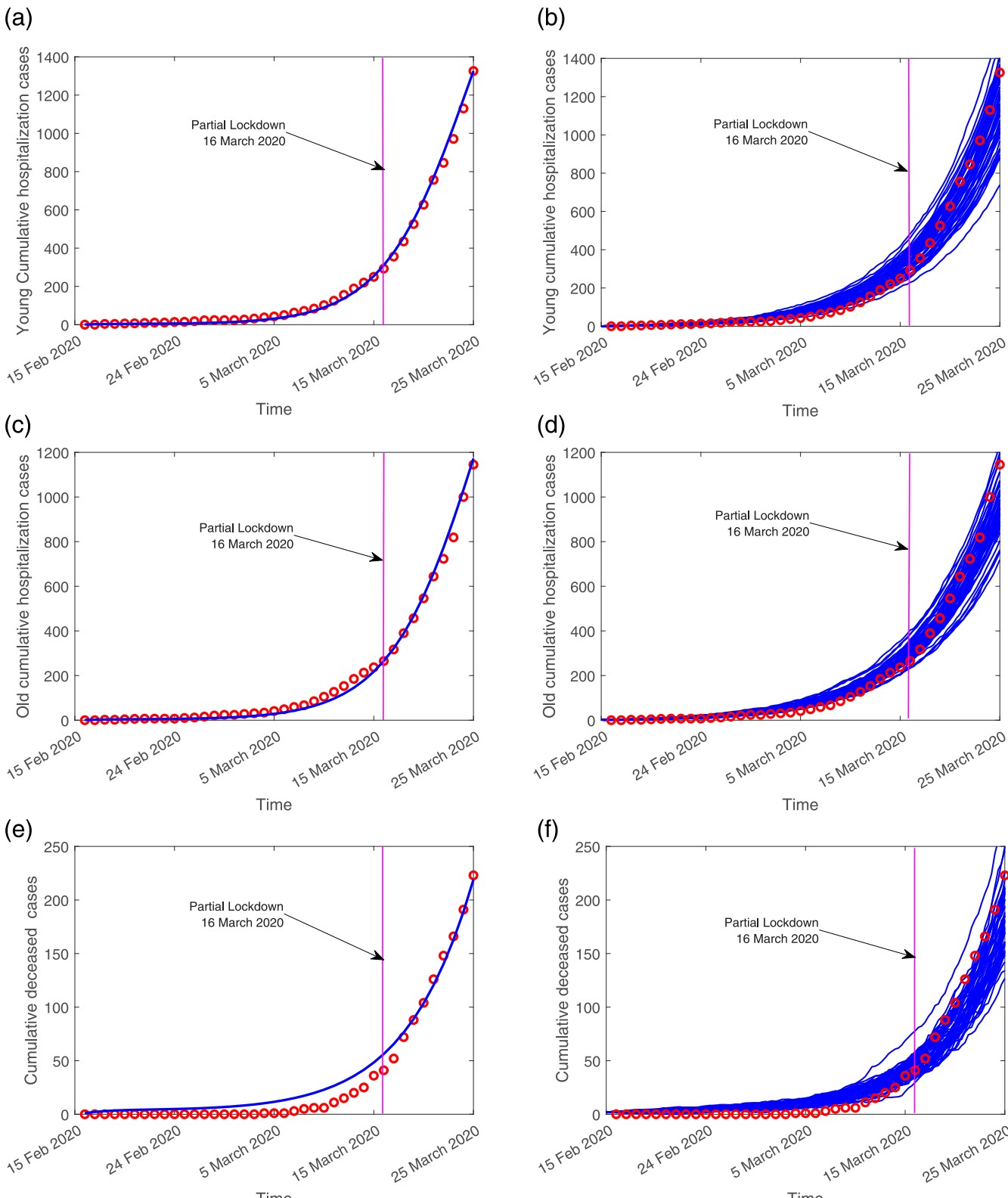

**Fig 6. On the left hand side, the deterministic model curve (blue line) and on the right hand side, the stochastic model realizations (in blue), fitting the cumulative empirical data (red dots).** In (a-b) the hospitalizations for the young group (0–69 years), in (c-d) the cumulative hospitalizations for the old group (70 years and older) and in (e-f) overall deceased cases.

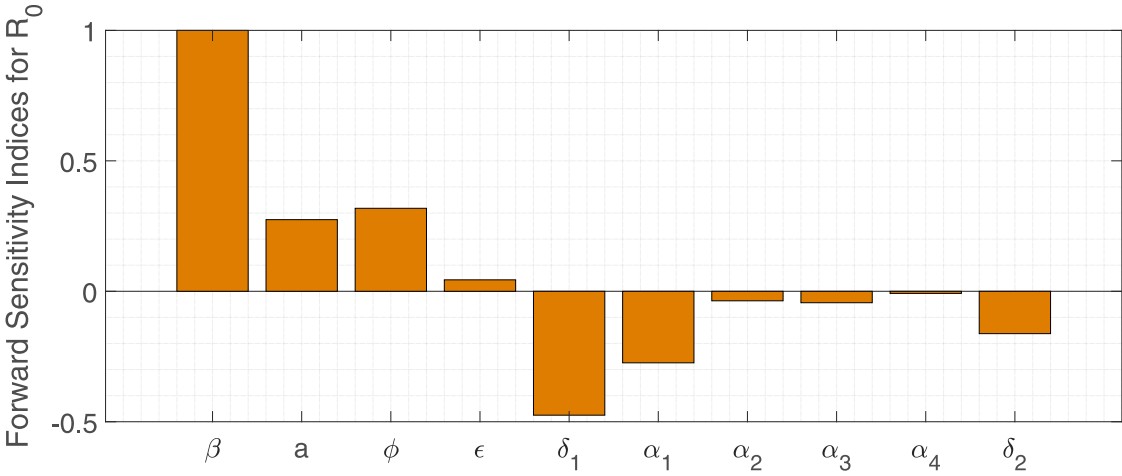

**Fig 7. Normalized forward sensitivity indices of $R_0$.**

## 4.2 Model simulations: An exploratory analysis

In this section, we explore different parameter combinations for the disease infectivity factors $\phi$ and $\epsilon$ and for the disease induced mortality rate $\delta$ that are able to explain the exponential phase of the COVID-19 epidemic in the Basque Country. For both, the deterministic and stochastic models, the assumed biological parameters for COVID-19 dynamics were estimated for the normalized data, see Table 4. While for the deterministic model simulations we have

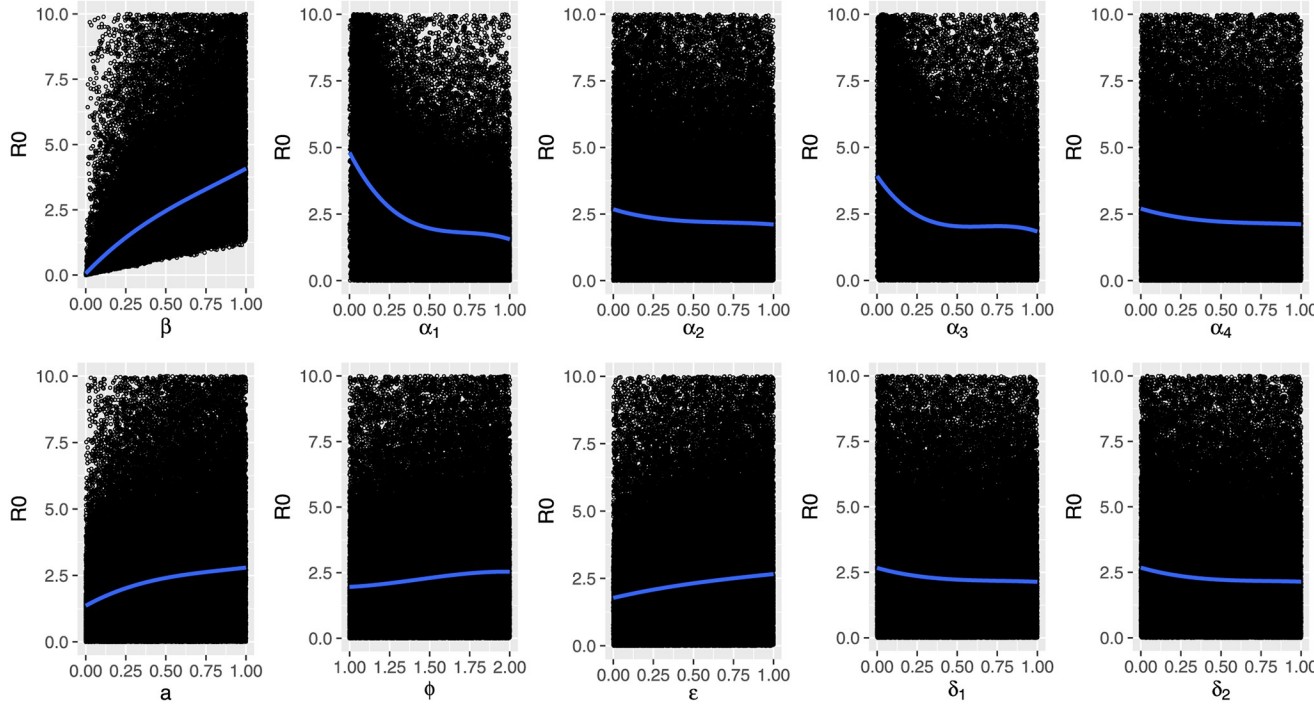

**Fig 8. Spline regression method to quantify the effect of the model parameter variation on $R_0$ behaviour.**

used the function ode45 in MATLAB, for the stochastic model simulations we have obtained 100 realizations using the Euler-Maruyama approach.

As an exploratory exercise to understand the impact of the key parameters on disease severity dynamics during the initial phase of the pandemics, numerical simulations are performed to describe the available data in the Basque Country, from February 15 to March 25, 2020, see Fig 10a). This is a dynamic work. While the present analysis focus on the introductory phase of the pathogen in the Basque Country, the evaluation of the effect of the imposed control measures will be carried out later.

Epidemiological data used in this study are provided by the Basque Health Department and the Basque Health Service (Osakidetza), continually collected with specific inclusion and exclusion criteria. We use the following incidence and cumulative data for RT-PCR (reverse transcriptase-polymerase chain reaction), see Fig 9. While the incidence data are shown in Fig 9a), the cumulative data used here refer to the overall hospital admissions, including the ICU cases, are shown in orange and the decease cases in black in Fig 9b).

Within the timeline of first wave of the pandemic in the Basque Country, the black line shows the date of the partial lockdown implementation, followed by the full lockdown, see red line. The light blue line shows the last data point used in this study, March 25, 2020, ten days after the partial lockdown was implemented, when the exponential growth of disease cases decelerates into a growth close to zero towards a linear phase [19].

To investigate the possible dynamics of hospitalizations for the young ($H_1$) and for the old ($H_2$) groups, as well as the dynamics for the overall deceased cases when no control measure would have been implemented in the Basque Country, a 100 days simulation time is shown, from February 15 to May 25, 2020, covering the post-lockdown period. The effects of different parameter combinations of the scaling factors of disease transmission and the disease induced mortality rates are shown in Fig 10. For the hospital admission cases dynamics, we evaluate the effect of the scaling factors affecting the disease transmission individually. By fixing $\phi = 1.5$ as estimated from the normalized data, we vary the value of the $\epsilon$ parameter, see Fig 10a) and 10b), while in Fig 10c and 10d) we fixed $\epsilon = 1.3$, varying the value of the $\phi$ parameter. The same experiment was performed for the deceased cases, see Fig 10e), varying the combination of the disease induced mortality $\delta$, always assuming $\delta_1 < \delta_2$.

Without any control measure, the epidemic would follow its course with a massive number of hospitalizations and deaths within the first 100 days of the pandemic. While a qualitatively similar dynamical behavior is observed when varying those key parameters, with an increase on the number of disease cases as the parameter value increases, the scaling factor $\phi$, differentiating the transmission between the mild/asymptomatic and the severe/hospitalized individuals, appears to affect significantly the older population, eventually reaching its maximum towards stationary, much faster than the dynamics in the young population.

This effect is also observed for the overall infection cases ($A_1 + H_1 + A_2 + H_2$), and for overall hospitalizations ($H_1 + H_2$), see Fig 11. Using both modeling approaches, deterministic and stochastic, our results have shown that disease cases would have eventually reached stationarity after 100 days if no control measure was implemented.

## 5 Discussion

Declared a pandemic by the World Health Organization (WHO) in March 2020 [2], the collective behavior of societies has been significantly affected by the extreme measures implemented to control disease transmission. As the COVID-19 pandemic progressed, research on mathematical modeling became imperative and very influential to understand the epidemiological dynamics of disease spreading and control under different scenarios. The hypothesis of a new

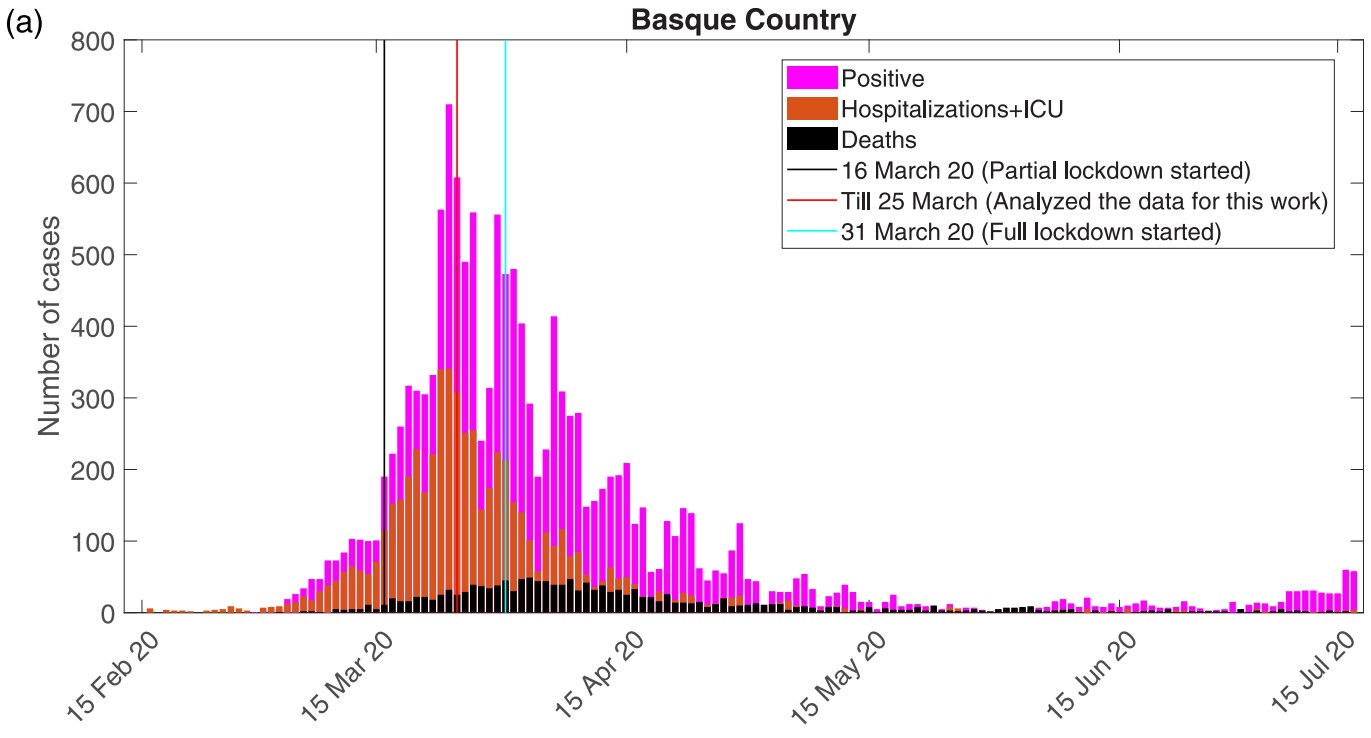

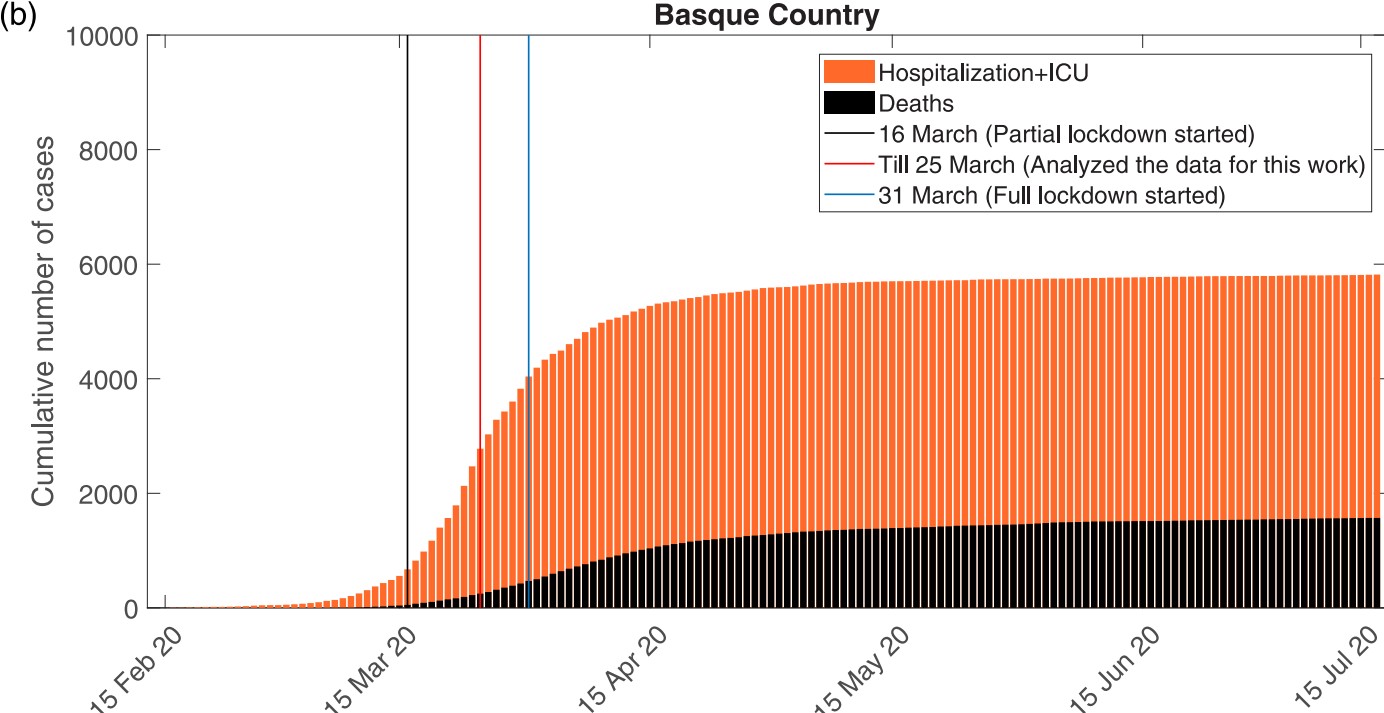

**Fig 9. COVID-19 epidemiological data in the Basque Country.** In (a) the cumulative hospital admissions and deceased cases. In (b) incidences for disease cases referring to hospitalizations including ICU and deaths.

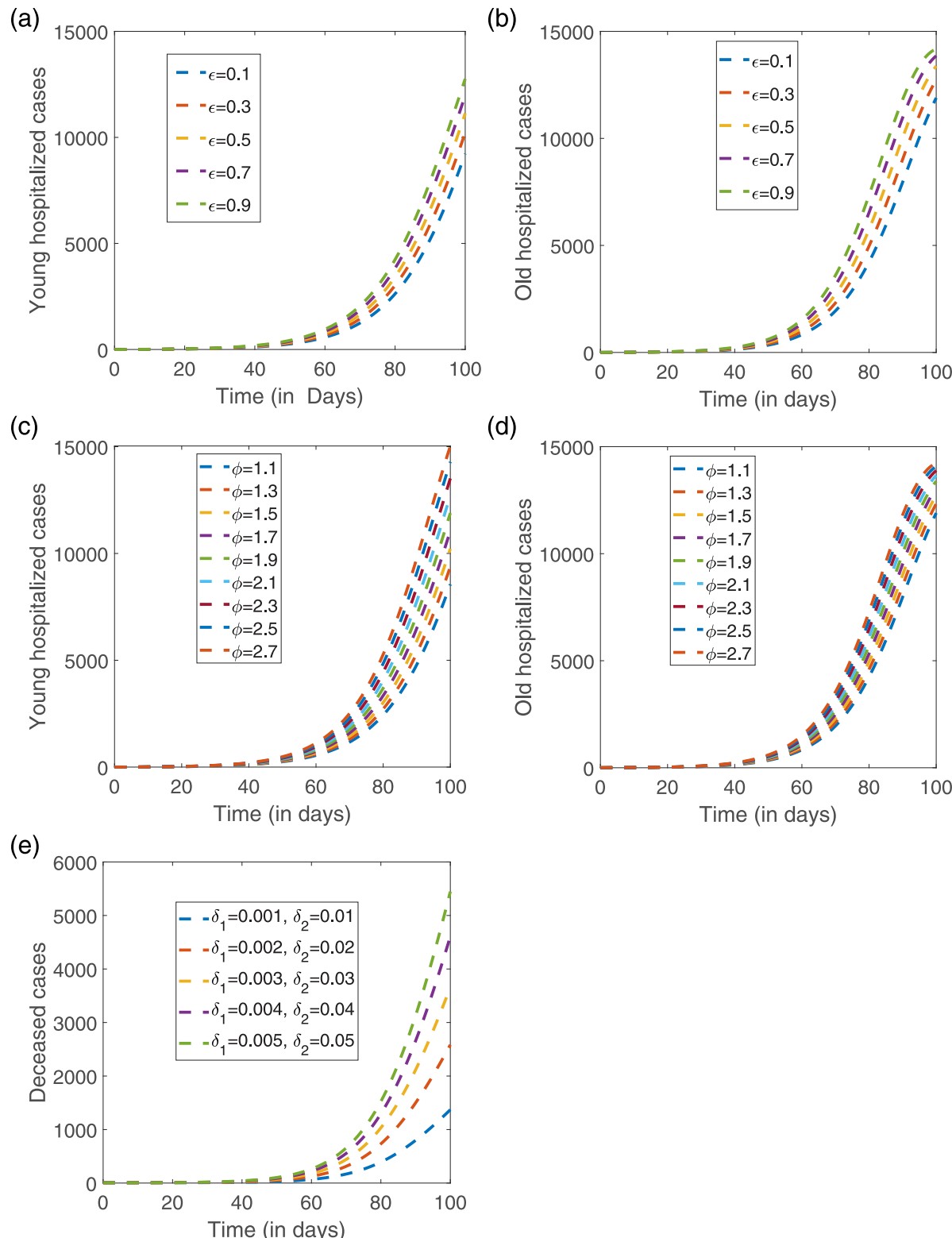

**Fig 10. Deterministic model simulations.**

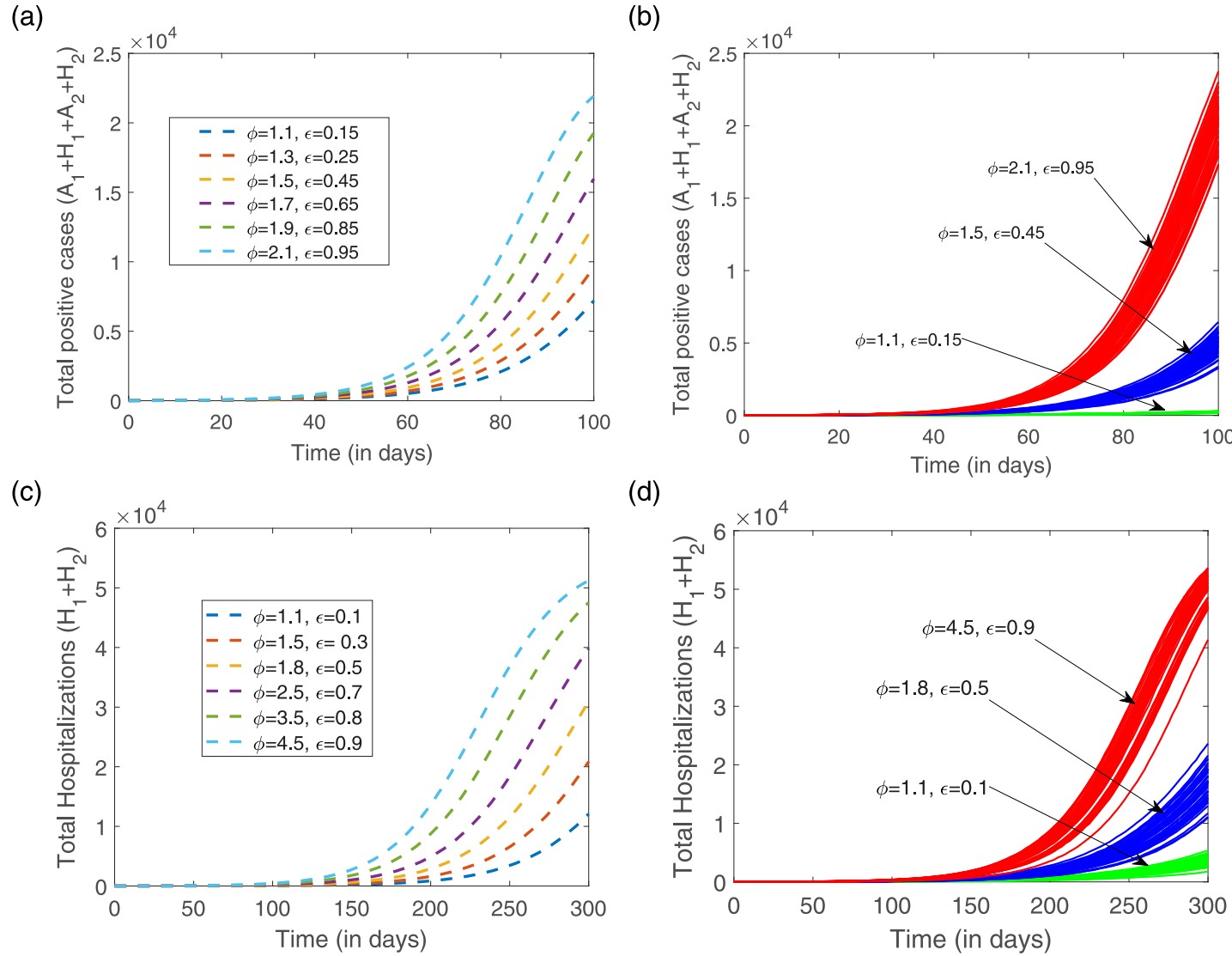

**Fig 11. By varying the infectivity scaling factors $\phi$ and $\epsilon$, the dynamics of the overall disease cases $(A_1 + H_1 + A_2 + H_2)$, and the dynamics of the overall hospitalization $(H_1 + H_2)$ are plotted for 100 and 300 days respectively, using the following parameter set: $\beta = 0.15$, $\delta_1 = 0.003$, $\delta_2 = 0.04$, $\eta_1 = 0.035$, $\eta_2 = 0.03$, $\alpha_1 = 0.02$, $\alpha_2 = 0.01$, $\alpha_3 = 0.05$, $\alpha_4 = 0.03$ and $a = 0.02$.** In (a) and (c) the deterministic model simulations and in (b) and (d) 100 stochastic realizations.

pathogen able to cause a very severe disease with an extremely high transmission rate were gradually adjusted overtime. It is now accepted that COVID-19 disease severity and death occur according to a hierarchy of risks, with age and pre-existing health conditions enhancing risks of disease severity.

In this paper, a mathematical model framework for COVID-19 transmission is proposed. Applied to the first wave of COVID-19 epidemic in Basque country, Spain, we stratify the population into young and old groups, after a detailed data analysis for the available epidemiological data referring to confirmed positive cases, hospitalization and deceased cases. The deterministic and the stochastic models are analyzed and results are compared.

For the deterministic approach, we calculate the disease-free equilibrium and the basic reproduction number ($R_0$). We show that disease-free equilibrium is global asymptotically

stable. A detailed sensitivity analysis is performed to identify the key parameters influencing the basic reproduction number, and hence, regulating the transmission dynamics of COVID-19.

Further, the deterministic model was extended to its stochastic counterpart. The stochastic differential equation (SDE) model is derived from a diffusion process. Simulations were obtained by the Euler-Maruyama method. Model derivation is shown in the S1 File.

Both models were able to fit well the empirical data using similar parameter value range, with the stochastic model always presenting a better result. A detailed sensitivity analysis was performed allowing us to identify the key parameters affecting the disease dynamics.

An exploratory analysis to understand the impact of those key parameters on disease severity dynamics during the initial phase of the pandemics, from February 15 to March 25, 2020, was performed. Numerical simulations have demonstrated that differences in infectivity from severe/hospitalized cases and mild/asymptomatic cases are the most important factors influencing the disease spreading in the population and without any control measure, the epidemic would have followed its course with a massive number of hospitalizations and deaths within the first 100 days of the pandemic.

These results are of use to guide public health authorities on disease control. The sensitivity analysis results shown in Figs 7 and 8 give insights on how to control the disease outbreak,

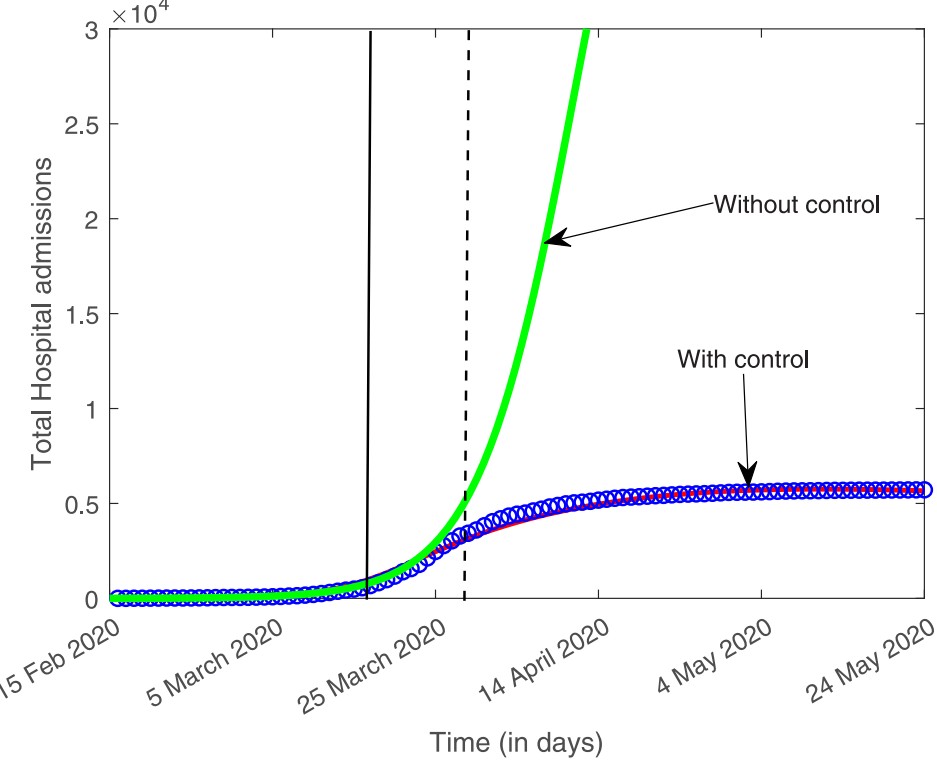

**Fig 12. The following parameter set: $\phi$ = 1.4, $\epsilon$ = 0.25, $\delta_1$ = 0.003, $\delta_2$ = 0.04, $\eta_1$ = 0.035, $\eta_2$ = 0.03, $\alpha_1$ = 0.02, $\alpha_2$ = 0.01, $\alpha_3$ = 0.05, $\alpha_4$ = 0.03 and $a$ = 0.02, the deterministic dynamics for the overall hospitalizations ($H_1 + H_2$) is shown with and without control.** Cumulative data on overall hospitalizations are shown in blue. The simulation plotted as red line includes a control function ($\beta(t) = \beta_0 \, \sigma_-(x(t)) + \beta_1 \, \sigma_+(x(t))$), with a standard sigmoid function $\sigma(x) = \frac{1}{1+e^{-x}}$, see [17]) which is able to describe the empirical data, while the green line shows the solution without any control. The black line shows the last data point used in this study, March 25, 2020, ten days after the partial lockdown was implemented. By that date, the exponential growth of disease cases decelerates into a growth close to zero towards a linear phase. The full lockdown started on March 31, 2020 (black dashed line).

suggesting possible ways of action for an effective containment of the disease transmission towards its elimination by limiting the increase of parameters with positive indices. On the other hand, by increasing the parameters with positive indices, such as providing treatment for a fast recovery or decreasing mortality, for example.

The numerical simulations have shown that without the lockdown, disease cases would increase continuously with severe cases eventually reaching its maximum numbers towards the herd immunity scenario, i.e, when a large portion of the population become immune to the disease. This behaviour was observed to affect the old group population much faster than the young population. Minimizing the scaling transmission factors, $\phi$ and $\epsilon$, via social distancing or vaccination, for example, would significantly reduce the disease burden in the population. Therefore, in terms of policy implications, our findings support the vaccination strategy prioritising the most vulnerable individuals to reduce hospitalization and deaths, as well as the non-pharmaceutical intervention measures, e.g social distancing and use of masks, that are still advised by the public health authorities, to reduce disease transmission.

This is a dynamic work. While the present analysis has focused on the initial phase of the COVID-19 epidemic in the Basque Country, it is important to mention that the evaluation of the effect of the imposed lockdown and other control measures is ongoing. As continuation of this work, the models are under refinement, using this framework as baseline to describe the progression of COVID-19 epidemics in the Basque Country and to understand the impact of lockdown implementation and the increased of testing capacity over time. As our model is able to describe the available data, see Fig 12, we will be also able to measure the impact of mild/asymptomatic cases on disease spreading and control, including non-sterilizing vaccine performance [21].

## Supporting information

**S1 File. Supporting information includes computation of the basic reproduction number ($R_0$) (S1 Appendix), proof of theorem 1.2 (S2 Appendix), stochastic process of the SIRS model (S3 Appendix), stochastic process of the proposed deterministic model 1 (S4 Appendix).**
(PDF)

## Acknowledgments

We thank Eduardo Millán, for collecting and preparing extended data sets on COVID-19 in the Basque Country. We thank Bruno Guerrero, research technician at the MTB group, for the support during the figures preparation.

## Author Contributions

**Conceptualization:** Akhil Kumar Srivasrav, Maíra Aguiar.

**Data curation:** Akhil Kumar Srivasrav, Joseba Bidaurrazaga Van-Dierdonck, Maíra Aguiar.

**Formal analysis:** Akhil Kumar Srivasrav, Maíra Aguiar.

**Funding acquisition:** Maíra Aguiar.

**Investigation:** Akhil Kumar Srivasrav, Nico Stollenwerk, Maíra Aguiar.

**Methodology:** Akhil Kumar Srivasrav, Nico Stollenwerk, Maíra Aguiar.

**Project administration:** Maíra Aguiar.

**Supervision:** Maíra Aguiar.

**Validation:** Joseba Bidaurrazaga Van-Dierdonck, Javier Mar, Oliver Ibarrondo, Maíra Aguiar.

**Writing – original draft:** Akhil Kumar Srivasrav, Nico Stollenwerk, Joseba Bidaurrazaga Van-Dierdonck, Javier Mar, Oliver Ibarrondo, Maíra Aguiar.

**Writing – review & editing:** Akhil Kumar Srivasrav, Nico Stollenwerk, Joseba Bidaurrazaga Van-Dierdonck, Javier Mar, Oliver Ibarrondo, Maíra Aguiar.

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
