## [Decision Letter · Decision Letter 0]

8 Jun 2022

PONE-D-22-11062Modeling the initial phase of COVID-19 epidemic: The role of age and disease severity in the Basque Country, SpainPLOS ONE

Dear Dr. Aguiar,

Thank you for submitting your manuscript to PLOS ONE. After careful consideration, we feel that it has merit but does not fully meet PLOS ONE’s publication criteria as it currently stands. Therefore, we invite you to submit a revised version of the manuscript that addresses the points raised during the review process.

Please correct the minor issues that the reviewers pointed out.

We look forward to receiving your revised manuscript.

Kind regards,

Lucinda Shen

Staff Editor 

on behalf of 

**Constantinos Siettos**

**Academic Editor**

**PLOS ONE**

Journal Requirements:

Additional Editor Comments (if provided):

Reviewers' comments:

Reviewer's Responses to Questions

**Comments to the Author**

1. Is the manuscript technically sound, and do the data support the conclusions?

Reviewer #1: Yes

Reviewer #2: Yes

2. Has the statistical analysis been performed appropriately and rigorously? 

Reviewer #1: Yes

Reviewer #2: Yes

3. Have the authors made all data underlying the findings in their manuscript fully available?

Reviewer #1: Yes

Reviewer #2: Yes

4. Is the manuscript presented in an intelligible fashion and written in standard English?

Reviewer #1: Yes

Reviewer #2: Yes

5. Review Comments to the Author

Reviewer #1: The manuscript deals with a determinitic stage structured model along with corresponding stochastic extension. Here authors have chosen SEIHR model for both young and old populations and derived their interaction pattern. Interestingly they found that the younger population may be able to affect the disease spread which eventually cause fatal impact to older population. This is a preliminary study on initial phase of the pandemic. The work is reasonably good and may be acccepted for publications.

There are a few corrections **(mostly typos)** which authors should take cere of. For author's help I have mentioned a few.

1. In abstract they mentioned 'younger than 60' but in line 400 it is 'younger than 40' for young population

2. In section 1.1 Ei etc should be defined as E1, E2 etc for better understanding as there are only two classes.

3. Theorem 1.1 and 1.2, components of E0 should be 10. Pl remove extra zeros.

4. Line 140, replace prove by state.

5. Expression after line 150, correct the term as $X_8(t)$

6. Line 169, write 600,000 instead of 600 thousand

7. Line 197, it should be pharmaceutical not non-pharmaceutical

8, In caption of figure 3 and 6, replace left by right and right by left

9. Caption Fig 12, write "The following .....". Also check expression of $\\beta(t)$, it is incorrect.

10. Line 386 ans 387: SIRS in place of SIR

11. Expression before line 390: O(\\Delta t) in place of O(t).

12. Line 405, \\sigma seems undefined. Pls check.

Reviewer #2: This manuscript deals with modeling COVID-19. The mathematical model is formulated by dividing the whole population under consideration into two different age groups, namely, older and younger keeping in view that the rates of transmission, hospitalization and recovery may vary for individuals in these two groups. The basic reproduction number (R0) is computed. The key parameters of the proposed model are estimated using nonlinear least square method based on data from the Basque Country, Spain. Sensitivity analysis is performed to see the impact of different parameters related to infectivity, recovery rates etc. The proposed model is extended to stochastic model and simulation results of both deterministic and stochastic models are compared. Overall manuscript is well written and mathematically sound. However, authors are requested to take a note of the following minor points and should make the necessary changes in the final manuscript:

1. Line no 202: "is" should be changed to "are".

2. Caption of Figure 6: left hand side is deterministic and right hand side is stochastic.

3. Caption of Figure 11: "In a-c)" should be "In (a) and (c)". Similarly, "b-d)" should be "(b) and (d)".

4. Caption of Figure 12: "the ollowing" should be "following". Check the control function included in this caption. x(t) is appearing with positive and negative signs.

5. Line 365 and 273: "dynamic" seems more appropriate in place of "dynamical".

6. PLOS authors have the option to publish the peer review history of their article (what does this mean?). If published, this will include your full peer review and any attached files.

Reviewer #1: No

Reviewer #2: No

---

## [Author Response · Author response to Decision Letter 0]

13 Jun 2022

A pdf file with the detailed answer to the reviewers is uploaded.

---

## [Editor Report · Decision Letter 1]

17 Jun 2022

Modeling the initial phase of COVID-19 epidemic: The role of age and disease severity in the Basque Country, Spain

PONE-D-22-11062R1

Dear Dr. Aguiar,

We’re pleased to inform you that your manuscript has been judged scientifically suitable for publication and will be formally accepted for publication once it meets all outstanding technical requirements.

Kind regards,

Constantinos Siettos, Ph.D.

Academic Editor

PLOS ONE
---

## [Editor Report · Acceptance letter]

22 Jun 2022

PONE-D-22-11062R1 

Modeling the initial phase of COVID-19 epidemic: The role of age and disease severity in the Basque Country, Spain 

Dear Dr. Aguiar:

I'm pleased to inform you that your manuscript has been deemed suitable for publication in PLOS ONE. Congratulations! Your manuscript is now with our production department. 

Kind regards, 

on behalf of

Professor Constantinos Siettos 

Academic Editor

PLOS ONE